# Implementation-independent Knowledge Graph Construction Workflows using FnO Composition

Gertjan De Mulder [ORCID] and Ben De Meester [ORCID]

IDLab, Department of Electronics and Information Systems,
Ghent University – imec, Technologiepark-Zwijnaarde 122, 9052 Ghent, Belgium
{firstname.lastname}@ugent.be

**Abstract.** Knowledge Graph construction is typically a task within larger workflows, with a tight coupling between the abstract workflow and its execution. Mapping languages increase interoperability and reproducibility of the mapping process, however, this should be extended over the entire Knowledge Graph construction workflow. In this paper, we introduce an interoperable and reproducible solution for defining Knowledge Graph construction workflows leveraging Semantic Web technologies. We describe how a data flow workflow can be described interoperable (i.e., independent from the underlying technology stack) and reproducible (i.e., with detailed provenance) by composing semantic abstract function descriptions; and how such a semantic workflow can be automatically executed across technology stacks. We demonstrate that composing functions using the Function Ontology allows for functional descriptions of entire workflows, automatically executable using a Function Ontology Handler implementation. The semantic descriptions allow for interoperable workflows, the alignment with P-PLAN and PROV-O allows for reproducibility, and the mapping to concrete implementations allows for automatic execution.

## 1 Introduction

Knowledge Graph (KG) construction – i.e., RDF graph construction – involves computational tasks on data, and is typically a task within larger (business or scientific) workflows. The construction of a KG itself can also be considered an overarching and more complex task that is composed of smaller tasks, e.g., extracting data from a database, mapping it to RDF, and publishing it using a web API (i.e., Extract-Transform-Load or ETL). Such a process – i.e., a set of tasks that can be automated – can be facilitated using a workflow system.

When a tight coupling between the abstract workflow and its execution exists, interoperability diminishes and composing tasks into a workflow introduces challenges to connect tools that implement a task. Similar issues arise when integrating a KG construction task into a larger workflow. For example, connecting a mapping implemented in JAVA and a web API tool implemented in JavaScript.

Mapping languages increase interoperability and reproducibility of the mapping process, however, this should be extended to the entire KG construction workflow. The lack of interoperability inhibits use of different tools for a task, making it harder to adapt to changing requirements and constraints. For example, Tool A might initially suffice for the RDF-generation task given the size of the source data. Later on, the data size might become unmanageable for Tool A. Tool B is available that can handle larger data sets, however, the lack of interoperability prevents the flexibility in switching from one tool to the other.

In this paper, we represent tasks within a workflow through the composition of implementation-independent semantic function descriptions. By providing interoperability between tasks and the tools that execute them, users can focus on the overarching task for which the workflow was created, for example, managing the KG construction life cycle using different mapping processors that generate RDF, and different endpoints on which the RDF is published.

Section 2 presents related work. In Section 3, we show how interoperability between tasks and tools within a workflow can be achieved through the composition of declarative function descriptions. We showcase this in Section 4 by leveraging the Function Ontology (FnO) [7] to obtain a data flow workflow that is decoupled from the tools that are used, therefore, illustrating the flexibility in choosing the technology to be used for each task. In Section 5, we demonstrate the resulting workflow composition in FnO. We conclude in Section 6 and give additional pointers for future work.

## 2   Related work

In this section, we discuss existing RDF graph construction workflows, and workflow systems' interoperability and reproducibility characteristics.

Compared to scripting, using a mapping language improves interoperability of the KG construction process [6]. Mapping languages can provide features to cover many steps within the KG construction process, i.e., not only specify how to map to RDF, but also how to extract data from different data sources [8], and how to publish using various methods [16]. Even when mapping languages provide enough features to be deemed end-to-end, executing a KG construction exists within a wider context, e.g., being part of a Knowledge Graph Lifecycle [4], or as a collection of subtasks to allow for optimization [13]. As such, even though KG construction rules can be described interoperably using, e.g., a mapping language, its position within the wider and narrower tasks makes it interpretable as being (a part of) a workflow.

Flexible workflows are needed, as requirements and constraints are subject to change. Thus, interoperability is essential for tasks designed in one system to be used by another [14]. The state of the art puts forward following characteristics for interoperability: 1) declarative paradigm, 2) separation of description and implementation, and 3) standardized language.

Statements within an imperative paradigm are exact instructions of what needs to be done and inherently define the control flow: the exact order in which a

program must be executed. An imperative paradigm is suitable for processes that are unlikely to change, however, a **declarative approach is recommended** when workflows resemble processes with changing requirements and constraints that require them to be executed in different ways. Declarative paradigms can be used to represent data flow, i.e., the data dependencies between tasks, and are more robust to change as they describe what needs to be done, instead of how [1].

Interoperability diminishes when there is a tight coupling between tasks and implementations [12], e.g., when using ad hoc approaches. Thus, the **separation of description and implementation** is crucial to interoperability [15].

The **use of standards** is essential to achieve interoperability in heterogeneous environments. Several workflow specifications exist, and can be divided into two parts. On the hand, there are executable specifications, such as the Common Workflow Language (CWL), and on the other hand, descriptive specifications, such as P-PLAN, and Open Provenance Model for Workflows (OPMW). CWL allows for describing a computational workflow and the command-line tools used for executing its tasks [3], with a tight coupling between tasks and implementations. P-PLAN extends the W3C standard PROV. It allows for describing workflow steps and link them to execution traces, and was applied in projects that focus on interoperability [10] and reproducibility [11]. OPMW is an extension of P-PLAN [10]: a simple interchange format for representing workflows at different levels of granularity (ie. abstract model, instances, executions). These specifications are either focused on being executable or descriptive. To the best of our knowledge, however, no specification exists that supports both.

The Function Ontology (FnO) [7] presents a similar approach towards interoperable data transformations using Semantic Web technologies. An implementation-independent function description allows for a decoupled architecture that separates the definition from its execution, and the inputs and outputs of a function are explicitly described. Furthermore, a recent update to FnO includes composition: compose a new function from other functions.

Reproducibility is another key characteristic within workflows, as it requires the tasks to be described in sufficient detail so that it can be reproduced in different environments [11]. In order to be reproducible by other scientists, **provenance information including the execution details** is required [2].

## 3   Method and Implementation

In this paper we put forward our approach towards interoperable and reproducible workflows through implementation-independent and declarative descriptions, allowing the flexibility of tasks being implemented by different tools. We discussed several existing description languages for defining workflows. The complexity of the language increases with the constructs that are supported. However, it appears that simplicity often pays greater dividends when considering interoperability. In that regard, we decided to look for lightweight – yet flexible and interoperable – solutions.

The previous section shows that to have interoperable and reproducible workflow, we need a declarative paradigm that separates description from implementation in a standardized language, and allows for generating provenance information for individual tasks. In this section we elaborate on the decisions that were made to accommodate for these characteristics.

We represent a workflow as a composition of tasks, and a task as a function which can have zero or more inputs and zero or more outputs. Being uniquely identifiable and unambiguously defined increases the reusability of tasks across workflows, as they are universally discoverable and linkable [7].

We make the simplification that tasks can only be executed sequentially and currently do not consider control flow constructs other than a sequence. The data flow between tasks within a composition is represented by input and output mappings between functions. Such a composition mapping describes how an input or output of one function is linked to the input or output of another function. For example, within a KG construction workflow this is needed to connect the output of an RDF generation task to the input of the subsequent publishing task.

We consider the Function Ontology (FnO) as a model to describe functions and function compositions to represent tasks and workflows. Its simple model aligns with our goal without preventing us to add additional complexity such as mapping to concrete implementations and composition of functions. Both additions are part of the Function Ontology specification[1].

The addition of composition to the FnO specification allows us to align function compositions with workflows as defined in P-PLAN [9], complementary to the existing alignment between FnO and PROV-O [5]. Several related works used or extended P-PLAN and led to the creation of several applications. Consequently, by aligning with P-PLAN we benefit from existing work that provides interoperability with several prominent workflow systems [10]. We use FnO because it allows for linking functions to actual implementations, hence, providing sufficient detail to be directly executed.

Therefore, by mapping the workflows defined as function compositions, to workflow descriptions in P-PLAN, we can benefit from those applications, such as the workflow mining, browsing, and provenance visualization solutions discussed in [10]

The following shows how FnO and P-PLAN align, and Listing 1.1 shows how construct P-PLAN descriptions from FnO compositions:

- fno:Execution is-a p-plan:Step
- fnoc:Composition is-a p-plan:Plan
- fno:Parameter is-a p-plan:Variable
- fno:Output is-a p-plan:Variable
- fno:expects is-a p-plan:isInputVarOf
- fno:returns is-a p-plan:isOutputVarOf

---

[1] https://w3id.org/function/spec/

```
1    PREFIX p-plan: <http://purl.org/net/p-plan#>
2    PREFIX fnoc: <https://w3id.org/function/vocabulary/composition#>
3    PREFIX fno: <https://w3id.org/function/ontology#>
4    PREFIX rdf: <http://www.w3.org/1999/02/22-rdf-syntax-ns#>
5
6    CONSTRUCT {
7      ?s a p-plan:Plan .
8      ?exX a p-plan:Step ; p-plan:isStepOfPlan ?s .
9      ?exY a p-plan:Step ; p-plan:isStepOfPlan ?s ; p-plan:isPrecededBy ?exX .
10   }
11   WHERE  {
12     ?s rdf:type fnoc:Composition ;
13       fnoc:composedOf [ fnoc:mapFrom [ fnoc:constituentFunction ?fx ;
14                                        fnoc:functionOutput ?fxOut ] ;
15                         fnoc:mapTo   [ fnoc:constituentFunction ?fy ;
16                                        fnoc:functionParameter ?fyParameter ] ] .
17     ?exX fno:executes ?fx .
18     ?exY fno:executes ?fy .
19   }
```

**Listing 1.1.** Pseudo-SPARQL query for constructing the precedence relations in P-PLAN from the CompositionMappings in FnO.

## 4 Use case

In this section we discuss POSH (Predictive Optimized Supply Chain): a motivating use case showcasing the need for an interoperable KG construction workflow.

POSH is an imec.icon research project in which methods and software solutions are researched that leverage data to optimize integrated procurement and inventory management strategies. A data integration and quality framework is deemed necessary to increase the accuracy and reliability of supply chain data that has been collected from heterogeneous data sources (suppliers, customers, service providers, etc.). Within POSH, we developed a semantically-enhanced knowledge integration framework that uses various data repositories and external (meta)data to provide a clear overview of the current state of the supply chain and the necessary inputs for the prediction, optimization and decision support methods.

To this end, a KG is generated from the heterogeneous supply chain data and consequently exposed through a triple store endpoint. This enables our partners to take advantage of running queries against a uniform data model without being burdened with heterogeneous sources from which it constitutes, and focus on the designing algorithms for optimizing the supply chain. However, not all data was made available from the start but rather added progressively, and the requirements together with the mappings rules that satisfy them changed in parallel. Hence, the KG generation tasks need to be executed iteratively to incorporate the changes, which can become time-consuming when done manually. To iteratively accommodate for changing requirements and constraints, an implementation-independent workflow system was needed. Within POSH, we applied our method to provide workflow system flexibly enough to adapt to different technology stacks.

## 5   Demonstration

In this section we demonstrate a working example of an ETL workflow comprising two tasks: i) generating RDF; and ii) publishing the generated RDF. Due to space restrictions only excerpts of the descriptions are shown.

First, we define the task of generating RDF as a function that takes the URI to a mapping, and the URI to which the result should be written. We make use of the RML mapping language to have an interoperable RDF generation step. Secondly, we define the publishing task as a function which takes the URI to the generated RDF data as input parameter and outputs a URI to the endpoint through which it is published. These descriptions are shown in Listing 1.2.

```
1    @prefix fno: <https://w3id.org/function/ontology#> .
2    @prefix fns: <http://example.com/functions#> .
3
4    fns:generateRDF a fno:Function ;
5        fno:expects ( fns:fpathMappingParameter ) ; fno:returns ( fns:returnOutput ) .
6
7    fns:publish a fno:Function ;
8        fno:expects ( fns:inputRDFParameter ) ;      fno:returns ( fns:returnOutput ) .
```
**Listing 1.2.** Task descriptions in FnO

We describe an overarching ETL task as the composition of these two functions, illustrated in Listing 1.3. We define how the data flows between the composed functions using fnoc:CompositionMapping. fnoc:Composition links the output of the first task to the second task by means of a fnoc:CompositionMapping. Note that, using composition, we are able to describe the workflow at multiple levels of abstraction. In analogy with an ETL workflow, for example, the highest level of abstraction represents the three Extract, Transform, and Load tasks. The second level can contain more specific, yet abstract, tasks that are required to fulfill each of the three Extract, Transform, and Load tasks. Depending on the complexity of each task, it can be described further in a lower level of abstraction.

```
1    @prefix fno: <https://w3id.org/function/ontology#> .
2    @prefix fnoc: <https://w3id.org/function/vocabulary/composition#> .
3    @prefix fns: <http://example.com/functions#> .
4
5    fns:ETL a fno:Function ;
6        fno:expects ( fns:fpathMappingParameter fns:fpathOutputParameter ) ; fno:returns ( fns:returnOutput ) .
7
8    fns:ETLComposition a fnoc:Composition ;
9        fnoc:composedOf
10        [ fnoc:mapFrom [ fnoc:constituentFunction fns:ETL ;
11                         fnoc:functionParameter fns:fpathMappingParameter ] ;
12          fnoc:mapTo   [ fnoc:constituentFunction fns:generateRDF ;
13                         fnoc:functionParameter fns:fpathMappingParameter ] ] ,
14        [ fnoc:mapFrom [ fnoc:constituentFunction fns:ETL ;
15                         fnoc:functionParameter fns:fpathOutputParameter ] ;
16          fnoc:mapTo   [ fnoc:constituentFunction fns:generateRDF ;
17                         fnoc:functionParameter fns:fpathOutputParameter ] ] ,
18        [ fnoc:mapFrom [ fnoc:constituentFunction fns:generateRDF ;
19                         fnoc:functionOutput fns:returnOutput ] ;
20          fnoc:mapTo   [ fnoc:constituentFunction fns:publish ;
21                         fnoc:functionParameter fns:inputRDFParameter ] ] ,
22        [ fnoc:mapFrom [ fnoc:constituentFunction fns:publish ;
23                         fnoc:functionOutput fns:returnOutput ] ;
24          fnoc:mapTo   [ fnoc:constituentFunction fns:ETL ;
```

```
25                              fnoc:functionOutput fns:returnOutput ] ] .
```
**Listing 1.3.** ETL Workflow description using FnO composition

We created a proof-of-concept Function Handler that automatically executes these descriptions using different implementations, available at https://github.com/FnOio/function-handler-js/tree/kgc-etl. Furthermore, we provide tests[2] in which we verify the execution sequence of a function composition, and demonstrate the interoperability through function compositions that resemble a KG construction workflow in which the RDF-generation task can be implemented by different tools.

## 6  Conclusion

Declarative function descriptions, and compositions thereof, allow us to define workflows that are decoupled from the execution environment. The explicit semantics allow for the unambiguous definition of inputs, outputs and implementations. Hence, allowing for automatically determine the functions that can be used to execute a task. Alignment with PROV allows for a reproducible workflow as both tasks and execution details are provided, which enables to exactly determine which functions were applied throughout the execution of the workflow.

Defining a workflow through compositions allows for different levels of abstractions. When rapid prototyping is required, only high-level tasks can be described. As requirements become more concrete, a high-level task can be described in greater detail as a composition of more fine-grained tasks.

These various levels of abstraction also allows for various levels of provenance information and thus various levels of reproducibility. For example, at one end of the spectrum, a function can be implemented by a command-line tool: no provenance information is available about the transformations that have been applied to produce the output. At the other end of the spectrum, a task can be described as a (nested) composition of fine-grained functions: provenance information is available up to the level of atomic functions.

For future work, we can see a mapping language as a way to describe compositions of transformation tasks. By representing, e.g., a Triples Map in RML as a composition of data and schema transformation tasks, we can provide insights in what a mapping does, and in what order. These insights could help to provide optimization strategies to such kind of engines.

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
