# OpenReview forum: "Implementation-independent Knowledge Graph Construction Workflows using FnO Composition"
_kg-construct.github.io/KGCW/2022/Workshop — KGCW 2022_

### Official Review · ~Herminio_García-González1 · 2022-03-24
**Promising idea but not well presented and argumented**

**Rating:** 5
**Confidence:** 4

**Review:**

This paper explores the definition of a KG generation workflow by means of the composition of abstract functions (using FnO). This composition provides an interoperable and reproducible framework in which the actual implementation could be more flexible to business changes. The problem tackled in this paper is a really relevant one. Achieving a successful solution would mean that KG workflows could be easily shared among practitioners and, therefore, the whole KG generation process would be more approachable and faster. That would finally redound in the amount of published LOD.

While, as mentioned, the topic is relevant and the idea is strong, I find the paper quite erratic and the arguments not very well connected. Moreover, it seems that the authors spend too much space giving arguments and motivation for the proposed solution but rather little space to really explain the potential of the proposal. In this regard, the given example is very little and does not give an intuition of the promised nested composition of functions. Namely, it is now not evident why one should opt for this solution instead of going for the one proposed in [16].

Apart from that, the reproducibility argument is mentioned a lot of times. However, as being abstract and implementation independent if I need to define a function that is is not already implemented I will have to share it. If not, the whole reproducibility would be hindered and consequently it will not work in other stacks. This issue, in my opinion, would really limit its feasibility in a bigger scale.

Following with this argument, then it is not clear why it is more interesting to define the workflows in this way, instead of integrating these capabilities to a standard language from which it would be possible to share the workflows without the necessity to share the implementation (moreover, without taking care of the implementation).

Then, the authors mention that is possible to create nested composed functions in order to really add fine grained detail on how a function behaves. This sounds as the same premisses used by the functional paradigm. Unfortunately, this is neither mentioned nor cited anywhere. Moreover, being able to define a function as a chain of higher-order functions (e.g., map, filter and reduce) would really mean the possibility to solve the need to share the implementation details for the shake of reproducibility.

As a side note, the authors put a lot of emphasis in the alignment with P-PLAN. Then in Listing 1.1 a possible alignment is demonstrated by means of a SPARQL Construct query. However, many selected variables are lost in the conversion (e.g., ?fx, ?fy). So, it is not clear why this alignment is interesting as it seems that we are losing a lot of expressiveness.

Finally, the related work section reads more like the continuation of the motivation part in the introduction. I would have expected a real discussion and comparison with other existing solutions. For example, how it compares to [16] or how it would surpass the use of Apache NiFi as proposed in [4].

Some other comments and typos per section:

#Introduction

Knowledge Graph (KG) construction - i.e., RDF graph construction - involves → Knowledge Graph (KG) construction involves

using a web API (i.e., Extract-Transform-Load or ETL) → using a web API, which fits under the Extract-Transform-Load (ETL) framework

Such a process [...] can be facilitated using a workflow system. (Rephrase)

the lack of interoperability prevents the flexibility in switching from one tool to the other. (This problem could be tackled defining mapping
rules translations which is approached by another paper submitted to this workshop.)

Section 2... → The rest of the paper is structured as follows: Section 2...

#Related work

On the hand → On one hand

An implementation-independent (margin overflow)

FnO includes composition: compose a new function (in general it is not a good idea to define a term using the same term)

that simplicity often pays greater dividends (It seems to me that this is not the most adequate expression. In addition, this a strong claim, it should be supported by a reference).

#Conclusion

These various levels of abstraction also allows for → These various levels of abstraction also allow for

#References

[15] (should be et al. in English)

---

### Official Review · ~Julián_Arenas-Guerrero1 · 2022-03-26
**Relevant topic, but could be more clear**

**Rating:** 6
**Confidence:** 3

**Review:**

This paper addresses knowledge graph generation using workflows in order to enhance the interoperability of these pipelines. Although I found the topic relevant and interesting for the KGC community, I think that the work could have been better presented, specifically making it more concise, and with richer examples. Also I think a brief introduction to P-PLAN or PROV-O could make the paper more self-contained.

In section 2 it is highlighted that standards are important for interoperability, however this paper proposes the use of FnO, which is not a standard nor widely adopted (yet).

It is not explained in detail why this proposal is needed for the presented use case or the benefits w.r.t. other approaches. One disadvantage of this declarative approach is the need of the implementation of the function handler, which is an additional component in the pipeline to maintain. A discussion of this would be interesting.

Overall the proposal is relevant and interesting but could have been more concise and better presented.

Minor typo:
Page 4: how TO construct

---

### Official Review · ~Maria-Esther_Vidal2 · 2022-03-30
**Good application of the convergence of mapping rules and the functional programming  paradigm, unfortunately, it is not very well presented**

**Rating:** 6
**Confidence:** 5

**Review:**

The paper addresses the problem of knowledge graph creation. The proposed pipeline resorts to the composition of functions expressed as FnO statements in RML to provide a two-step workflow that creates an RDF graph and loads the created graph into a triple store. The proposed approach is demonstrated in the context of a use case where supply chain data is integrated. The reported result puts in perspective the expressive power of RDF mapping languages and the functional programming paradigm.

The paper lacks a high-level description of the implemented workflow and the definition of the predicates utilized from existing vocabularies (e.g., P-PLAN and FnO). These omissions prevent a precise definition of the formal meaning of the presented workflow.
 Additionally, nothing is said about the scalability of the approach whenever the workflow is evaluated using existing RML-compliant engines.

All these issues reduce the value of the current version of this work. However, given the importance of the addressed topic, the recommendation is for acceptance. However, the authors are encouraged to reorganize the article and describe the workflow and the used predicates more precisely. Also, the discussion of the workflow implementation and execution will provide a
a complete picture of the pros and cons of the workflow implementation.

---

### Official Review · ~Pano_Maria1 · 2022-04-01
**Promising approach, but presentation needs work**

**Rating:** 6
**Confidence:** 3

**Review:**

The proposal in this paper is promising and would open up a range of possible applications.
The paper itself is, at points, hard to follow, and remains quite abstract.
An overview of the relevant concepts, preferrably also visualized, would help greatly in conveying the approach.

In section 3 a concrete example would help to understand the approach better.

In section 4 an example of how the accomodation of changing requirements would work in this approach again would provide clarity.

Additionally, I miss a discussion of relation between this approach and existing industry approaches for declarative ETL workflow management. How would they relate? Can all those features be captured by this approach?

The presentation of the approach needs some work, but definitely warrants discussion in this workshop.

---

### Decision · Program_Chairs · 2022-04-11

**Decision:**

Accept

**Comment:**

Dear authors,
Thank you for submitting your paper. We are happy to inform you that we accept your paper! Please carefully consider the reviews when you prepare your paper for the camera-ready version. You will receive specific instructions to submit your camera-ready soon.

We encourage the authors to read extremely careful the reviews and make the necessary changes in order to enhance the presentation of the paper's contributions for the camera-ready version. As reviewers mentioned, major changes are required.

Congratulations!